# Completely aqueous processable stimulus responsive organic room temperature phosphorescence materials with tunable afterglow color

Dan Li[1], Yujie Yang[1], Jie Yang [1✉], Manman Fang[1], Ben Zhong Tang[1,2✉] & Zhen Li [1,3,4,5✉]

Many luminescent stimuli responsive materials are based on fluorescence emission, while stimuli-responsive room temperature phosphorescent materials are less explored. Here, we show a kind of stimulus-responsive room temperature phosphorescence materials by the covalent linkage of phosphorescent chromophore of arylboronic acid and polymer matrix of poly(vinylalcohol). Attributed to the rigid environment offered from hydrogen bond and B-O covalent bond between arylboronic acid and poly(vinylalcohol), the yielded polymer film exhibits ultralong room temperature phosphorescence with lifetime of 2.43 s and phosphorescence quantum yield of 7.51%. Interestingly, the RTP property of this film is sensitive to the water and heat stimuli, because water could destroy the hydrogen bonds between adjacent poly(vinylalcohol) polymers, then changing the rigidity of this system. Furthermore, by introducing another two fluorescent dyes to this system, the color of afterglow with stimulus response effect could be adjusted from blue to green to orange through triplet-to-singlet Förster-resonance energy-transfer. Finally, due to the water/heat-sensitive, multicolor and completely aqueous processable feature for these three afterglow hybrids, they are successfully applied in multifunctional ink for anti-counterfeit, screen printing and fingerprint record.

[1] Institute of Molecular Aggregation Science, Tianjin University, 300072 Tianjin, China. [2] Shenzhen Institute of Molecular Aggregate Science and Engineering, School of Science and Engineering, The Chinese University of Hong Kong, Shenzhen, 2001 Longxiang Boulevard, Longgang District, 518172 Shenzhen City, Guangdong, China. [3] Tianjin Key Laboratory of Molecular Optoelectronic Sciences, Department of Chemistry, Tianjin University, 300072 Tianjin, China. [4] Department of Chemistry, Wuhan University, 430072 Wuhan, China. [5] Joint School of National University of Singapore, Tianjin University, International Campus of Tianjin University, 350207 Binhai New City, Fuzhou, China. ✉email: jieyang2018@tju.edu.cn; tangbenz@cuhk.edu.cn; lizhen@whu.edu.cn

There is a growing interest in stimulus-responsive luminescent materials which could undergo physical or chemical changes in response to the external stimuli, such as mechanical force, heat, light, and pH[1–6], due to the potential application of such materials in the fields of information storage, anti-fake, and optoelectronic devices[7–9]. To date, despite the ever-increasing variety of stimulus-responsive systems have been reported, most of the stimulus-responsive luminescent materials have been based on fluorescence[10–12]. As the result, the response could only be monitored from the changed emission color or intensity under the external stimulus. Therefore, it is necessary to develop stimulus-responsive materials from another dimension, such as emission lifetime, which could broaden their practical application in much more fields.

Organic room-temperature phosphorescence (RTP), one recently popularized phenomenon, has received significant attention over the past few years owing to the low toxicity of materials, long emission lifetimes, and large Stokes shifts to enable potential utility in numerous applications[13–18]. In particular, in comparison with fluorescent materials with short lifetime, RTP ones with a longer lifetime even caught by the naked eye are more conducive to their development as stimulus-responsive materials[19–24]. Nevertheless, the exploration of stimulus-responsive RTP materials is still at the preliminary stage. The main reasons can be summarized as follows: (i) the RTP emission tended to be realized in crystal, greatly limiting its applications; (ii) it was extremely difficult and complicated to simultaneously control triplet excitons and stimulus-response sites. In light of this, if external stimuli could destroy or rebuild the intermolecular interaction of RTP materials on the macro level, it will provide a simpler method for designing such materials.

Herein, we report a stimulus-responsive ultralong RTP material, namely DPP-BOH-PVA, which is sensitive to water and heat. In our previous work, the arylboronic acid ester, 1,1′:3′,1″-terphenyl-5′-boronic acid ester (DPP-BO), exhibits remarkable phosphorescent properties at low temperature or under mechanical stimulation[25]. Inspired by this, arylboronic acid, 1,1′:3′,1″-terphenyl-5′-boronic acid (abbreviated as DPP-BOH), which tends to occur dehydration condensation reaction for the existence of boronic acid unit[26], was chosen as a phosphorescent chromophore to react with poly(vinylalcohol) (PVA) polymer chains, to yield DPP-BOH-PVA (Fig. 1). There are two reasons for choosing PVA as the matrix. On one hand, the hydroxyl groups in PVA could react with the hydroxyl groups of DPP-BOH to form B–O covalent bonds, which could limit the thermal motion of DPP-BOH molecule, then facilitating RTP emission; on the other hand, PVA exhibits excellent hydroscopicity. The rigidity of PVA chains will be broken in a humid environment, which provides a stimulus-responsive site. Therefore, the resultant RTP property could be controlled by the alternating stimulation of heat and water. Furthermore, by introducing another two fluorescent dyes to this system, such as fluorescein and rhodamine B, the afterglow color could be adjusted through triplet-to-singlet Förster-resonance energy transfer (TS-FRET). At this time, the stimulus-response property of these systems could still be retained, as PVA matrix acted as the stimulus-responsive site. More importantly, the fabrication processes of these materials are only based in pure water phase without any organic solvents, which is environmentally friendly and adheres to the purpose of green chemistry. All the products in this work are accessible to be obtained, which may provide a new perspective for designing stimulus-responsive ultralong RTP materials.

## Results

**Stimulus-responsive room-temperature phosphorescence property of films.** As shown in Fig. 1, DPP-BOH-PVA film was prepared through a simple method of dehydration condensation reaction between DPP-BOH and PVA (mass ratio = 1:100) under the addition of ammonia water. The photophysical properties of desiccative DPP-BOH-PVA film were systematically investigated. The obtained film exhibited bluish-violet fluorescence at 345 nm under UV-light irradiation (254 nm) (Supplementary Fig. 1). Impressively, the blue phosphorescence with emission peak at 475 nm appeared when turning off the UV lamp and could be captured by the naked eye even lasting for 10 s (Fig. 2a, e and Supplementary Movie 1). The

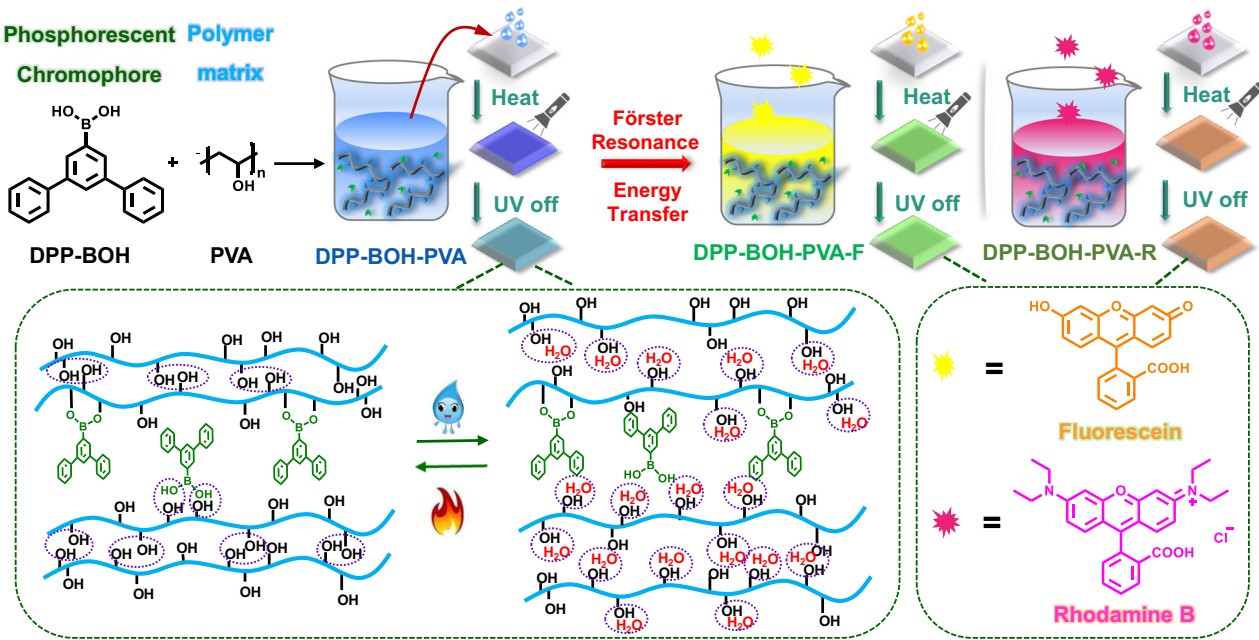

**Fig. 1 Stimulus-responsive room-temperature phosphorescent system.** Schematic illustration of the synthetic process of three target products and changes in intermolecular interactions under heating or water stimulus.

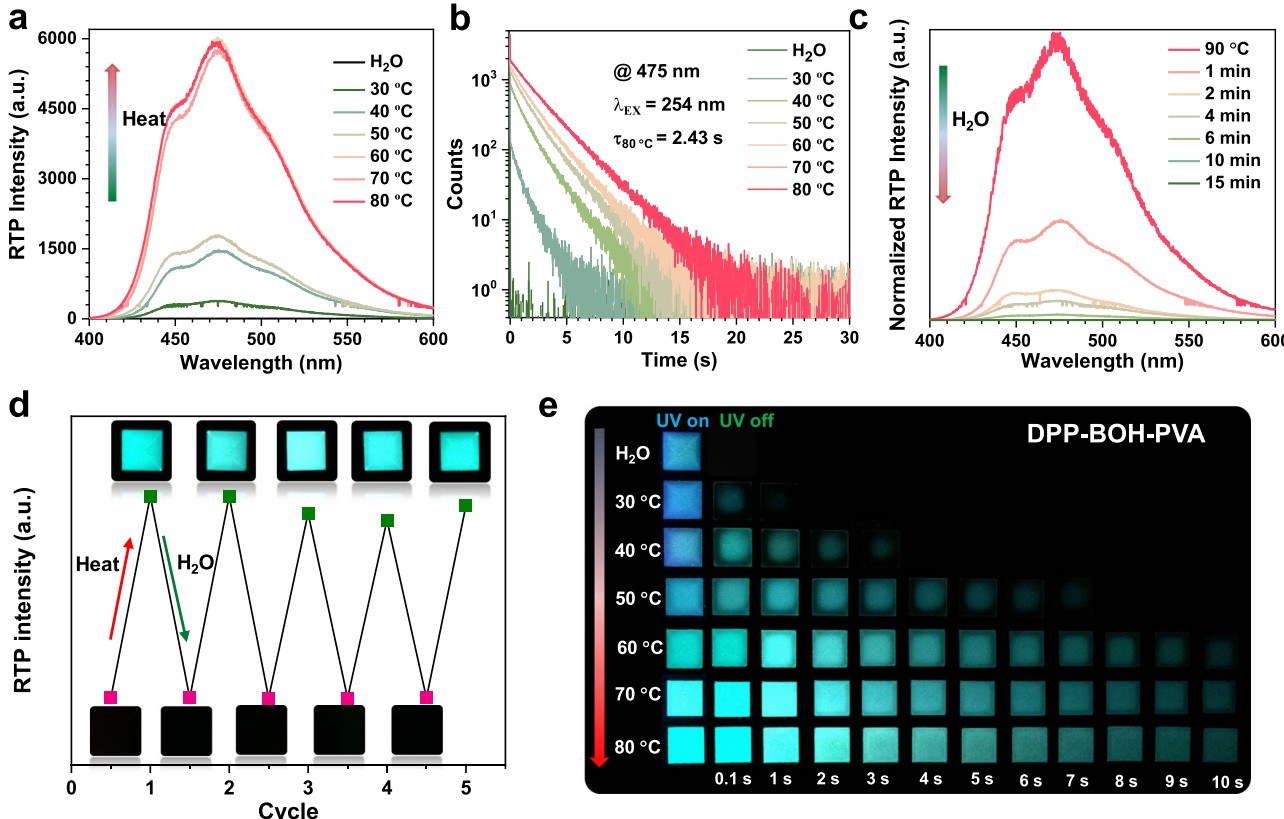

**Fig. 2 Photophysical properties of DPP-BOH-PVA film under the stimuli of water and heat. a** Phosphorescence spectra of water-fumed DPP-BOH-PVA film after heating at different temperatures for 15 min. **b** Time-resolved emission-decay profiles of water-fumed DPP-BOH-PVA film after heating at different temperatures for 15 min. **c** Phosphorescence spectra of desiccative DPP-BOH-PVA film under water fuming for different times. **d** Repeated cycles of the heating/water fuming processes and the corresponding photographs of DPP-BOH-PVA film after turning off the UV irradiation. **e** Photographs of water-fumed DPP-BOH-PVA film after heating at different temperatures (30 °C–80 °C). The temperature gradient was 10 °C and the corresponding heating time was 15 min, after which the RTP behaviors were studied when the samples were cooled to room temperature.

time-resolved emission-decay curve showed that the RTP lifetime of DPP-BOH-PVA reached up to 2.43 s, while the corresponding RTP quantum yield could achieve 7.51% (Fig. 2b and Supplementary Tables 1 and 2), surpassing most of the organic RTP materials under ambient condition.

To validate that water did have an effect on the RTP emission of DPP-BOH-PVA film, the RTP spectra and corresponding lifetimes of the film were measured after alternating stimulation of heat and water (Fig. 2). First, the film fumed by water vapor for 15 min was measured, which showed nearly no RTP emission at 475 nm (Fig. 2a). After confirming its good thermal stability (Supplementary Figs. 3 and 4 and Supplementary Table 3), the heating-responsive RTP effect was explored. As the heating temperature arose, the RTP intensity of the 475 nm emission band gradually enhanced. When the heating temperature was 60 °C, the RTP intensity reached a plateau and hardly enhanced even if elevating the heating temperature unceasingly. By this time, the water in the film should be considered to be removed almost. In addition, the RTP lifetime also exhibited a similar changing tendency. That was, the lifetime of RTP emission gradually prolonged as the heating temperature increased, reaching the maximum of 2.43 s at 80 °C (Fig. 2b).

Subsequently, we explored the reverse process of removing water by heating the film at 90 °C and then fuming with water vapor for different times. As shown in Fig. 2c, the RTP intensity of the 475 nm emission band gradually decreased with the extension of water fuming time. This further confirms that the water indeed affects the RTP property. Thus, the RTP property of

film could be controlled by heating and water fuming, and the cycle could be repeated many times (Fig. 2d and Supplementary Fig. 2).

**The mechanism for ultralong organic phosphorescence of DPP-BOH-PVA film.** In order to explore the mechanism for stimulus-responsive RTP effect of DPP-BOH-PVA film, DPP-BOH-PVA-C was prepared as a control. The only variable in the preparation process is that there is no addition of alkali to catalyze the reaction between DPP-BOH and PVA. As expected, DPP-BOH-PVA-C film shows unsatisfactory RTP performance with a lifetime of 0.48 s and a phosphorescent quantum yield of 2.86% even after heating. Analyzing the UV–vis absorption spectra of these two films (Fig. 3a), it could be found that DPP-BOH-PVA film shows an extra absorption peak at about 445 nm compared to DPP-BOH-PVA-C, which may be derived from the formation of B–O covalent bond between DPP-BOH and PVA. The reaction yield was proved to be 46.50% by UV absorption measurement (Supplementary Fig. 5). As for DPP-BOH-PVA-C, lacking this kind of covalent bond, the hydrogen-bonding interactions are not strong enough to build a rigid environment, so the RTP performance is much inferior to that of DPP-BOH-PVA. Even though, the RTP effect of DPP-BOH-PVA-C film is also sensitive to water (Supplementary Figs. 6–8 and Supplementary Table 4). Further on, DPP-BO (1,1′:3′,1″-terphenyl-5′-boronic acid ester), the analog of DPP-BOH, was prepared, in which the reaction site to PVA was protected by acid ester. When DPP-BO

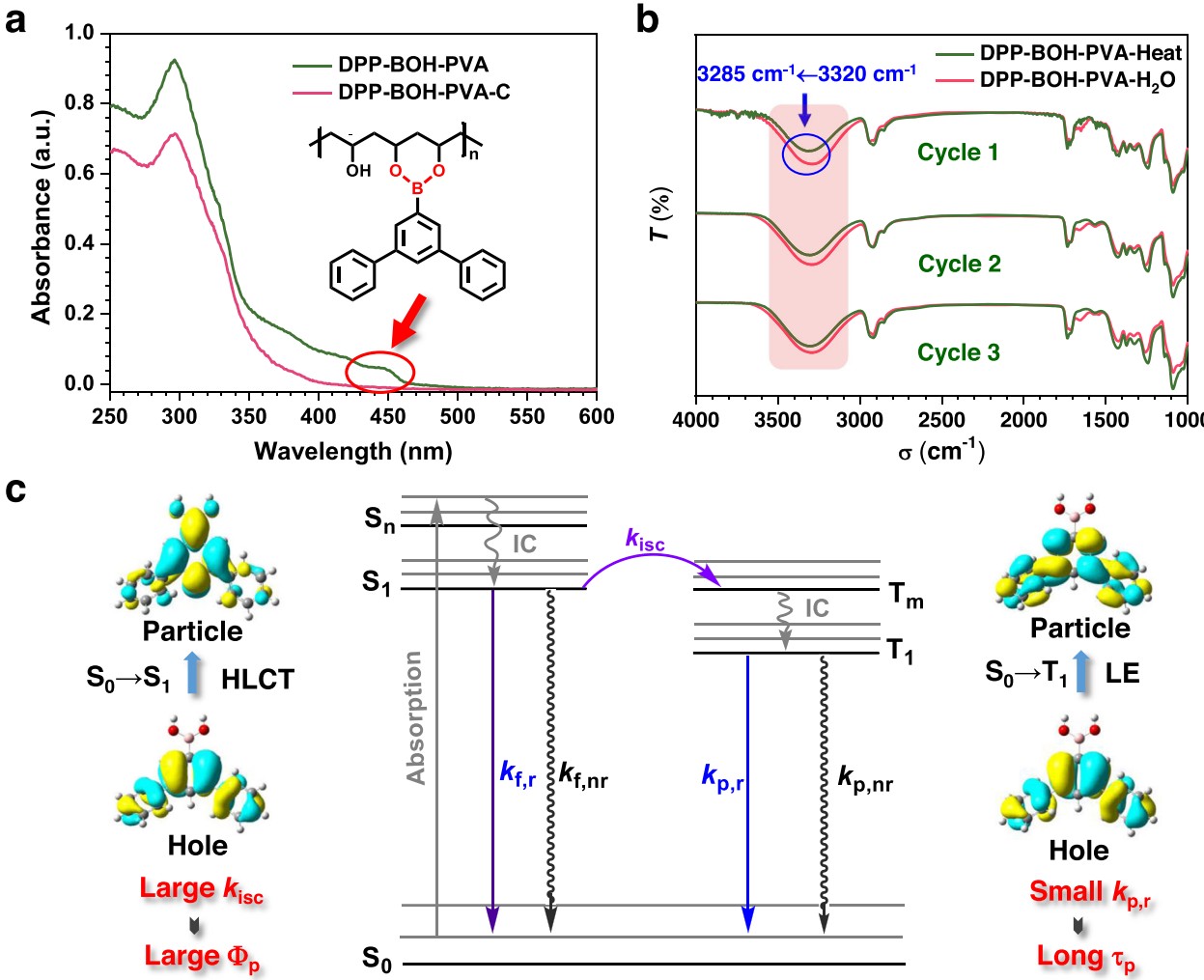

**Fig. 3 The mechanism for ultralong phosphorescence of DPP-BOH-PVA film. a** The absorption spectra of DPP-BOH-PVA film and DPP-BOH-PVA-C film, and the chemical structure of DPP-BOH-PVA. **b** Three repeated cycles for the Fourier transform infrared (FTIR) spectra of DPP-BOH-PVA film under of the heating/water stimuli. **c** The Jablonski diagram and theoretical calculations about natural transition orbitals (NTOs) for DPP-BOH (HLCT = hybridized local and charge transfer, LE = locally excited, IC = internal conversion, $k_{isc}$ = rate constant of intersystem crossing, $k_{p,r}$ = radiative rate constant of phosphorescence, $\Phi_p$ = phosphorescence quantum yield, $\tau_p$ = phosphorescence lifetime).

was mixed with PVA, nearly no RTP emission could be detected for its corresponding film even after heating, although DPP-BO could give strong blue phosphorescence with lifetime up to 4.40 s at low temperature (at 77 K) (Supplementary Figs. 9–12). These results demonstrate that the ultralong RTP property is largely related to the formation of B–O covalent bond between DPP-BOH and PVA, which could control the tightness of the molecules more effectively than hydrogen-bonding interactions. Moreover, the phosphorescence spectrum of DPP-BOH in the solution state at 77 K is consistent with the RTP spectrum of DPP-BOH-PVA film, suggesting that the phosphorescence emission source of DPP-BOH-PVA film is from DPP-BOH (Supplementary Fig. 12). In addition, there was almost no change in phosphorescence intensity of DPP-BOH-PVA film under oxygen atmosphere for 5 min. Even for 15 h, the oxygen did not completely quench the phosphorescence, indicating the close arrangement of molecules caused by hydrogen bond and B–O covalent bond makes it difficult for oxygen to enter (Supplementary Figs. 13 and 14).

Then, the theoretical calculations were carried out to understand the internal mechanism of remarkable RTP property from DPP-BOH-PVA film (Fig. 3c and Supplementary Figs. 15 and 16). The

natural transition orbitals (NTOs) of DPP-BOH were evaluated as shown in Fig. 3c. For the $S_1$ state, the hole distributes on the terphenyl units, while the particle almost spreads over the whole molecule skeleton. The partial overlap of the hole and the particle demonstrates a significant hybrid local and charge transfer (HLCT) character in the $S_1$ state, which facilitates the intersystem crossing (ISC) process from $S_1$ to $T_m$ and results in the larger ISC constant ($k_{isc}$). Thus, it contributes much to the resultant high phosphorescent quantum yield of DPP-BOH. Meanwhile, for the $T_1$ state, the hole and the particle both distribute on the terphenyl units. The absolutely overlap of the hole and the particle demonstrates a typical locally excited (LE) character, which makes the transition from $T_1$ to $S_0$ difficult and leads to a small spin–orbit coupling (SOC) value and phosphorescent radiative transition constant ($k_{p,r}$). Therefore, the phosphorescence lifetime ($\tau_p$) would be much prolonged based on the equation of $\tau_p = 1/(k_{p,r} + k_{p,nr})$. Besides, the $k_{isc}$ and $k_{p,r}$ for DPP-BOH-PVA film were calculated based on the experimental results. As shown in Supplementary Table 5, the $k_{isc}$ is as large as $6.34 \times 10^7$ s$^{-1}$, while $k_{p,r}$ and $k_{p,nr}$ are just 0.04 and 0.37 s$^{-1}$, which could well correspond to the theoretical results, and certify the accuracy of internal mechanisms mentioned above.

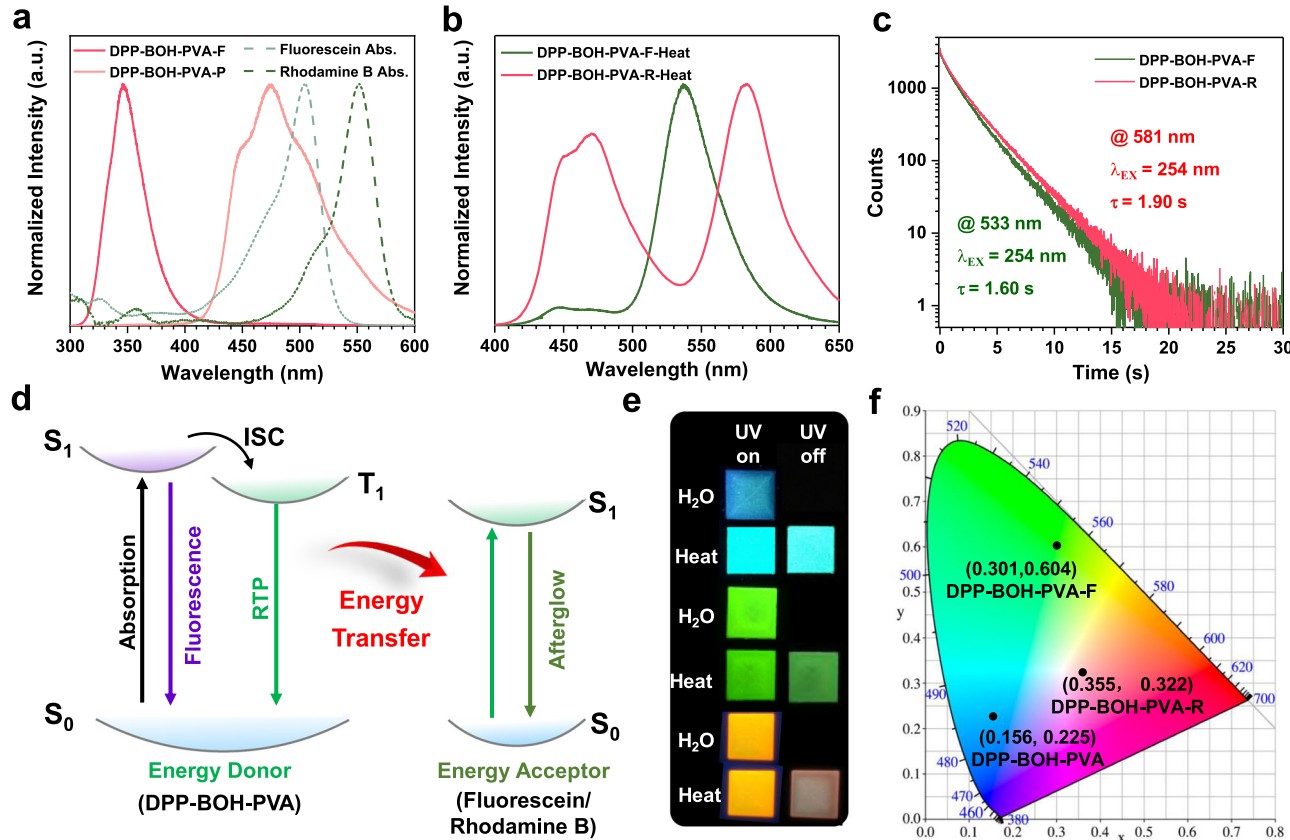

**Fig. 4 Tunable afterglow color through triplet-to-singlet Förster-resonance energy transfer (TS-FRET). a** The fluorescence/phosphorescence spectra of DPP-BOH-PVA and the UV–vis absorption spectra of fluorescein and rhodamine B. **b** The normalized RTP spectra of DPP-BOH-PVA-F and DPP-BOH-PVA-R. **c** Time-resolved emission-decay profiles of DPP-BOH-PVA-F (@ 533 nm) and DPP-BOH-PVA-R (@ 581 nm). **d** Simplified Jablonski diagram to explain the phosphorescence energy transfer (ISC = intersystem crossing, RTP = room-temperature phosphorescence). **e** RTP photographs of DPP-BOH-PVA, DPP-BOH-PVA-F, and DPP-BOH-PVA-R before and after heating at 90 °C for 15 min. **f** Commission Internationale de l'Eclairage (CIE) coordinates of afterglow emissions for DPP-BOH-PVA, DPP-BOH-PVA-F, and DPP-BOH-PVA-R.

To further explore the water-sensitive mechanism of DPP-BOH-PVA film, the Fourier transform infrared (FTIR) spectra were measured. As shown in Fig. 3b, the heated DPP-BOH-PVA films exhibit a noticeable peak at 3320 cm$^{-1}$, which should be ascribed to the associated hydroxyl group between adjacent DPP-BOH-PVA. When the film was fumed with water, the peak shape becomes wider and the peak shifts to short wavenumber of 3285 cm$^{-1}$, indicating that the presence of water in the film could increase the degree of association of hydroxyl groups. Combined with changes in phosphorescence properties, it is considered that the presence of water destroys the hydrogen-bonding interaction between adjacent DPP-BOH-PVA chains, resulting in the destruction of the rigid environment of this system, and thus the RTP emission was quenched. When the film was heated, the water was removed and intermolecular hydrogen bonds were constructed again, then recovering the RTP property. Therefore, the water-sensitive property of DPP-BOH-PVA film should be related to the destruction and reconstruction of intermolecular hydrogen bond interaction between adjacent DPP-BOH-PVA chains.

**Tunable afterglow color through triplet-to-singlet Förster-resonance energy transfer (TS-FRET).** In order to expand the stimulus-responsive afterglow materials, the afterglow color was further tuned. The strategy for adjusting the color of afterglow has been reported in 2020 by Subi J. George and co-workers, which utilized a long-lived phosphor as the energy donor and the fluorescent dyes as the energy acceptor to realize the triplet-to-singlet

Förster-resonance energy transfer (TS-FRET)[27–30]. Inspired by this, the commercially available fluorescent dyes, fluorescein, and rhodamine B were chosen as the energy acceptor in this work. As shown in Fig. 4a, the absorption spectra of fluorescein and rhodamine B show obviously spectral overlap with the phosphorescence spectrum of DPP-BOH-PVA film, which meets the prerequisite for energy transfer from triplet state to the singlet state. DPP-BOH-PVA-F and DPP-BOH-PVA-R were prepared by doping fluorescein and rhodamine B into DPP-BOH-PVA, which both exhibited multiple emissions (Supplementary Figs. 17–20). Particularly, two emission peaks could be still observed after turning off the 254 nm UV light (Fig. 4b). As seen from the afterglow spectrum of DPP-BOH-PVA-F film, the major emission peak locates at 533 nm from fluorescein and a weak one at about 475 nm from DPP-BOH-PVA, indicating the high energy transfer efficiency from DPP-BOH-PVA to fluorescein (Fig. 4d). As for DPP-BOH-PVA-R film, the intensity of two emission peaks at 581 nm (rhodamine B) and 475 nm (DPP-BOH-PVA) is almost equal, which is consistent with the phenomenon of the less spectral overlap between the absorption spectrum of rhodamine B and the phosphorescence spectrum of DPP-BOH-PVA (Fig. 4b). It is worth noting that DPP-BOH-PVA-F film and DPP-BOH-PVA-R film exhibit green and orange afterglows with lifetimes of 1.60 s (@ 533 nm) and 1.90 s (@ 581 nm) under ambient conditions, respectively (Fig. 4c). Certainly, their ultralong afterglows are also visible to the naked eye for about 10 s (Supplementary Movies 2 and 3). At this time, the emission lifetimes at 475 nm were measured to be 2.10 s and 2.20 s for DPP-BOH-PVA-F and DPP-BOH-PVA-R, respectively (Supplementary

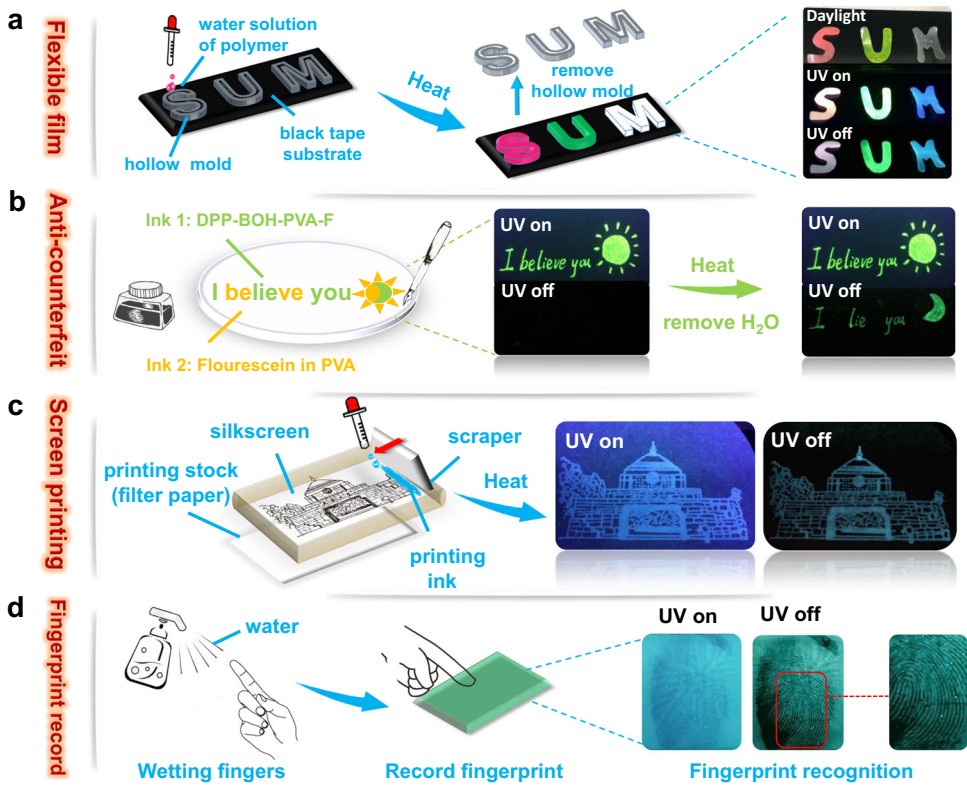

**Fig. 5 Applications of stimulus-responsive afterglow materials.** The schematic illustration of the application process for **a** flexible film, **b** anti-counterfeit, **c** screen printing, and **d** fingerprint record.

Fig. 19). Accordingly, the TS-FRET efficiencies ($\Phi_{FRET}$) could be calculated to be 13.58% and 9.47% for them based on the equation of $\Phi_{FRET} = 1 - \tau/\tau_0$, in which $\tau$ and $\tau_0$ are the RTP lifetimes of energy donor (DPP-BOH-PVA) after and before energy transfer (Supplementary Table 6)[31]. Besides, the rate constants of FRET from DPP-BOH-PVA to fluorescein and rhodamine B were calculated to be $0.06\,s^{-1}$ and $0.04\,s^{-1}$, respectively (Supplementary Table 6). As these data were larger or similar to the RTP radiative rate ($0.04\,s^{-1}$) of DPP-BOH-PVA, the FRET occurring from singlet to triplet state could be well certified. Furthermore, the oscillator strength (*f*) for the donor phosphorescence was calculated to be 0.0078 (Supplementary Table 5), which should be large enough to facilitate the dipole–dipole coupling with an acceptor.

Moreover, the films mixed with PVA and fluorescein or rhodamine B show almost no phosphorescence (Supplementary Fig. 21). Also no delayed emission signal could be detected with the optimal excitation wavelengths of fluorescein and rhodamine B for DPP-BOH-PVA-F and DPP-BOH-PVA-R (Supplementary Fig. 22). These exclude the afterglow originating from phosphorescence of fluorescein or rhodamine B. Excitedly, DPP-BOH-PVA-F and DPP-BOH-PVA-R films are sensitive to water, and the cycle by heating and water fuming could be repeated (Fig. 4e and Supplementary Figs. 23–30). In addition, the change of FTIR spectra is also similar to that of DPP-BOH-PVA film, which further confirms the mechanism proposed above (Supplementary Fig. 31). Therefore, a large range of color adjustment can be conveniently achieved for amorphous stimulus-responsive materials with ultralong afterglow, which is conducive to expand their practical applications in many fields (Supplementary Fig. 32).

**Applications of stimulus-responsive afterglow materials.** Taking advantage of the remarkable ambient multicolor afterglow and the water processability of these materials, four kinds of

potential applications were explored as follows. First, making full use of the flexibility of the polymer to prepare multicolor soft films with different letter shapes ("S", "U", and "M") is shown in Fig. 5a. When the UV lamp is turning on, these films show different colors of fluorescence, then afterglow appears after turning off the UV irradiation. In the future, it is practicable to choose more shapes of molds to prepare a variety of handicrafts.

Furthermore, the water processability of these materials allows them to be used as ink in many applications. For example, as shown in Fig. 5b, there are two kinds of inks prepared in this application, which both exhibit green fluorescence when the UV irradiation is on (254 nm). The difference is that one ink (ink 1) is the water solution of DPP-BOH-PVA-F, which shows green afterglow after heating when the UV irradiation is off. In the contrast, the other ink (ink 2) is the water solution of mixed fluorescein and PVA, which is non-emissive when the UV lamp is turned off even after heating. Then, these two inks are used to write different letters on the filter paper, respectively. Under 254 nm UV irradiation, a sentence of "I believe you (Sun)" with green emission could be observed. Due to the afterglow of DPP-BOH-PVA-F could be quenched by the presence of water before heating, nothing could be seen after turning off the UV lamp. After heating the filter paper for about 15 min, the water is removed, then the sentence of "I lie you (Moon)" appears after turning off the UV lamp, which means completely opposite to that when the UV light on. The stimulus-responsive materials with afterglow realize such a secondary anti-counterfeiting.

In addition, they also could be chosen as ink for silk-screen printing. Owing to the water solubility of these materials, they can directly use the filter paper as a substrate for printing, rather than limiting the choice of substrate material like most organic solvents. In Fig. 5c, DPP-BOH-PVA was used as the ink to print out the pattern of the gate of Tianjin University. Without heating,

nothing was displayed when the UV light was off, but after heating, the blue afterglow could appear.

Finally, based on the water-sensitive property of these materials, a fingerprint recording device was constructed (Fig. 5d). A desiccative DPP-BOH-PVA film could be prepared first. Then, wet your finger and press it on the prepared film for 5 s. The fingerprints could be clearly displayed when the UV lamp was turned off, because the protruding parts of the fingerprint wet the film, but the recessed position did not touch the film. This application simplifies the process of real-time fingerprint record collection and shows good practical application value.

## Discussion

In summary, a kind of stimulus-responsive ultralong RTP materials was designed and prepared. The fabricated DPP-BOH-PVA film shows the excellent RTP property. When the film is exposed to the water vapor, the RTP property disappears since the water can break the rigid environment in the system. Conversely, heating could remove water and recover the RTP property of the film. Meanwhile, with incorporating fluorescent dyes, the afterglow color was adjusted from blue to green to orange, through triplet-to-singlet Förster-resonance energy transfer. Finally, due to the highly water-processable property of these three afterglow hybrids, they show promising applications in multifunctional ink for anti-counterfeit, screen printing, and the simple fingerprint recording device. The development of the stimulus-responsive multicolor materials with ultralong afterglow will broaden multifunctional stimuli-responsive materials and expand applications in much more fields.

## Methods

**Reagents and materials**. Unless otherwise noted, all reagents used in the experiments were purchased from Jiangtian Chemical Co., LTD (Tianjin, China). The polyvinyl alcohol (PVA) was purchased from Macklin ($M_W \approx 20,000$, PDI: 1.16, alcoholysis degree: 87–89%). (3,5-diphenylphenyl)boronic acid (purity: 98%), fluorescein (purity: 97%), and rhodamine B (purity: biological stain, BS) were purchased from Heowns and used as received without further purification.

**Measurements**. UV–vis absorption spectra were obtained using a Shimadzu UV-2700. Steady-state photoluminescence/phosphorescence spectra and phosphorescence lifetime were measured using Hitachi F-4700. The fluorescence lifetime and photoluminescence quantum efficiency were obtained on FLS-1000. The luminescent photos were taken by iPhone 6 s under the irradiation of a hand-held UV lamp at room temperature.

**General procedure for the synthesis of three hybrids**. DPP-BOH-PVA film: To a stirred solution of PVA (500 mg) in water (3 mL), DPP-BOH (5 mg) in water (4 mL), and ammonium hydroxide (1 mL) was added. The mixture was stirred at 80 °C for 20 min. Subsequently, take 0.7 mL of the obtained aqueous solution and drop it on the cover glass with a syringe. Then, heat the cover glass until the water evaporates.

DPP-BOH-PVA-C film: To a stirred solution of PVA (500 mg) in water (3 mL), DPP-BOH (5 mg) in water (5 mL) was added. The mixture was stirred at 80 °C for 20 min. Subsequently, take 0.7 mL of the obtained aqueous solution and drop it on the cover glass with a syringe. Then, heat the cover glass until the water evaporates.

DPP-BO-PVA film: To a stirred solution of PVA (500 mg) in water (3 mL), DPP-BO (5 mg) in water (5 mL) was added. The mixture was stirred at 80 °C for 20 min. Subsequently, take 0.7 mL of the obtained aqueous solution and drop it on the cover glass with a syringe. Then, heat the cover glass until the water evaporates.

DPP-BOH-PVA-F film: To a stirred solution of PVA (500 mg) in water (3 mL), DPP-BOH (5 mg) in water (3 mL), ammonium hydroxide (1 mL), and fluorescein (0.5 mg) in water (1 mL) were added. The mixture was stirred at 80 °C for 20 min. Subsequently, take 0.7 mL of the obtained aqueous solution and drop it on the cover glass with a syringe. Then, heat the cover glass until the water evaporates.

DPP-BOH-PVA-R film: To a stirred solution of PVA (500 mg) in water (3 mL), DPP-BOH (5 mg) in water (3 mL), ammonium hydroxide (1 mL), and rhodamine B (0.5 mg) in water (1 mL) were added. The mixture was stirred at 80 °C for 20 min. Subsequently, take 0.7 mL of the obtained aqueous solution and drop it on the cover glass with a syringe. Then, heat the cover glass until the water evaporates.

**Theoretical calculation**. All density functional theory (DFT) calculations were performed using Gaussian 09 program. The ground state ($S_0$) structure and natural transition orbits (NTOs) of $S_1/T_1$ states for DPP-BOH were evaluated by the TD-m062×/6–31 g*.

## Data availability

The authors declare that the data supporting the findings of this study are provided in the Supplementary Information file. All data are available from the authors upon request.

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

## Acknowledgements

We are grateful to the National Natural Science Foundation of China (No. 51903188, J.Y.), the Natural Science Foundation of Tianjin City (No. 19JCQNJC04500, J.Y.), the starting Grants of Tianjin University and Tianjin Government (Z.L.), and the Independent Innovation Fund of Tianjin University (J.Y.) for financial support. We appreciate the Edinburgh Instruments fluorescence spectrophotometer (FLS-1000) for conducting fluorescence lifetime and quantum yield measurements.

## Author contributions

J.Y. and Z.L. conceived the project. B.T. gave valuable suggestions. D.L. was primarily responsible for the experiments, then measured and analyzed the optical data. Y.Y. and M.F. took the pictures. J.Y. conducted all the theoretical calculations. D.L., J.Y., and Z.L. wrote the manuscript.

## Competing interests

The authors declare no competing interests.
