## [Peer Review File · Nature Communications]

REVIEWER COMMENTS

Reviewer #1 (Remarks to the Author):

In this manuscript, the authors reported a kind of materials with room temperature phosphorescence emission by doping emissive molecules into polymer matrix. Such materials showed efficient RTP emission after heating treatment. However, I could not find any novelty and/or new fundamental insights from this manuscript. Meanwhile, several similar works have been reported recently. Considering the quality of this manuscript, I cannot recommend its publication in Nature Communications.

1. One of my major concerns of this manuscript is the novelty. The related luminescent mechanism of room temperature phosphorescence, applications and general strategy are almost identical to published works. (Sci. Adv. 2020, 6, eaaz6107; Angew. Chem. 2020, 132, 9479 –9483; ACS Appl. Mater. Interfaces 2020, 12, 20765–20774; Adv. Mater. 2020, 32, 1907355)
2. The DPP-BOH-PVA construction and the achievement of multiple color afterglow through energy transfer, which is the same as Ref (Sci. Adv. 2020, 6, eaaz6107 and Angew. Chem. 2020, 132, 9479 – 9483). Therefore, new fundamental insights could be found in this manuscript.
3. From the introduction, the authors stated that “it is necessary to develop ... another dimension”. It is so unclear to understand this, what is the dimension exactly? It may lead to some ambiguity for the reader.
4. From the Figure 2d, between the cycle of removal of water and heating, the phosphorescence color showed some obvious changes. For example, the color was green while turned to blue after one cycle of treatment, but it changed to green again after another cycle, which is significant in Supporting Figure 2. What are the internal reasons? Such problems also existed in the Figure 2e, fluorescence color changed under various temperatures. And Supporting Figure 4.
5. For the Figure 2d, after just one cycle treatment, the phosphorescence intensity decreased significantly. Whether the high temperature treatment would destroy the internal structures, which need to be verified by experiments.
6. The authors attributed the new absorption peak to the B-O covalent bond. However, no direct evidence can be derived from this manuscript to verify this. Some additional characterizations should be presented.
7. The proposed large ISC and low irradiative constant should be presented as well.
8. Such stimulus-responsive characteristics were just from the PVA, which can absorb water from atmosphere. Hence, such materials can be really called smart responsive materials? Additionally, the quenching of phosphorescence by water in PVA matrix have been reported in some works (ACS Appl. Mater. Interfaces 2020, 12, 20765–20774; Adv. Mater. 2020, 32, 1907355).
9. For the application as shown in Figure 5, the fingerprint displays were too vague to identify, which should be improved.

Reviewer #2 (Remarks to the Author):

The authors report a new organic room temperature phosphorescent material that responds to external stimuli. A blue room temperature phosphorescent material is obtained by the simple method of mixing poly(vinyl alcohol) (PVA) and a phosphorescent dye containing boronic acid in water and allowing it to dry. The importance of hydrogen bond formation in PVA for the emission of room temperature phosphorescence is investigated by several controlled experiments, and the addition of water breaks the hydrogen bonds, resulting in the switch-off of the phosphorescence. Demonstrations of multicolor phosphorescence and anti-counterfeiting applications show the usefulness of the current materials. Room temperature phosphorescent materials are a topic that has attracted a great deal of attention in recent years, and this simple method for creating stimuli-responsive, multi-color phosphorescence materials could be an important contribution. The following points should be considered before publishing this paper.

1. The NTOs of T_m involved in the inter-system crossing from S₁ to should be shown to see if there is any change in the shape of the orbitals.
2. The efficiency of ISC is desired to be determined.
3. For FRET from T₁ to fluorescein or rhodamine in DPP-BOH-PVA, the validity of FRET occurring from T₁ with a small transition moment should be discussed by calculating the rate constant of FRET and comparing it with the deactivation rate of the triplet.

Reviewer #3 (Remarks to the Author):

In the manuscript the authors have demonstrated stimulus-responsive cyan emissive room temperature phosphorescence from aryl boronic acid chromophores in PVA with an afterglow emission. Authors have explained that the formation of B-O covalent bond between phosphor and chromophores is responsible for the phosphor's increased rigidity, which results in a high phosphorescence lifetime. By taking advantage of the afterglow emission of the phosphors, authors have demonstrated green and orange afterglow emission from conventional fluorescence emitter through phosphorescence Förster resonance energy transfer (P-FRET/TS-FRET) process. On the other hand, very efficient phosphorescence emission in the low wavelength region with a lifetime of 2.43 s is another important aspect of this manuscript. Also, the authors have demonstrated multicolor security printing, which is beneficial for multilevel information encryption. The manuscript has enough novelty and is suitable to publish in this journal considering the recent interest in ambient organic phosphors. However, the author should address the following points in a revised manuscript, to quantify certain aspects which is important for a publication in a journal like Nature Communication.

1. To achieve high phosphorescence efficiency and lifetime, authors have heated the films to 80 °C. What will happen if authors go more than 80 °C? As I can see that emission is saturated at 60°C, but lifetime is still increasing.
2. Is there any experimental support that can quantitatively tell the conversion of B-O covalent bond formation.
3. Cyan emissive persistent phosphorescence emission is one of the exciting parts of this manuscript; authors should cite similar recent work, which is very relevant for this manuscript Adv. Funct. Mater. 2020, 30, 2003693.
4. In the Supplementary Table1, I can see no decrease in the donor lifetime at 30 °C, 60 °C, and 70 °C with acceptor undoped and doped samples. Is FRET is not operating here?
5. To get the delayed fluorescence lifetime from Fluorescein and Rhodamine B, authors have monitored the emission wavelength at 533 nm 581 nm; however, a significant donor contribution is observed. How did the author nullify the contribution from the donor lifetime in the delayed fluorescence lifetime of fluorescein and rhodamine B?
6. It is exciting to know whether the acceptors' emission amplification is there or not. On the other hand, I will suggest the authors show the acceptor's emission spectra upon direct excitation of the acceptors.
7. I will also suggest that the authors calculate the singlet to singlet FRET efficiency as I can see the decreased donor fluorescence lifetime data from 12.99 ns in the undoped dye sample (supplementary Figure 1) to 11.32 ns 11.76 ns in the fluorescein and rhodamine B (supplementary Figure 13) doped sample.
8. Other than the spectral overlap of the donor emission and the acceptor absorption, significant oscillator strength for the donor phosphorescence is required to facilitate the dipole-dipole coupling with an acceptor. The author should add more discussion about it.
9. Authors should cite the very relevant and seminal research works on TS-FRET process in the phosphorescence energy transfer: J. Phys. Chem. Lett. 2019, 10, 310–315, 2. Sci. Adv. 2019, 5, eaaw5978, Angew. Chem. Int. Ed. 2021, 60, 19720-19724.

REVIEWER COMMENTS

Reviewer #1 (Remarks to the Author):

In this manuscript, the authors reported a kind of materials with room temperature phosphorescence emission by doping emissive molecules into polymer matrix. Such materials showed efficient RTP emission after heating treatment. However, I could not find any novelty and/or new fundamental insights from this manuscript. Meanwhile, several similar works have been reported recently. Considering the quality of this manuscript, I cannot recommend its publication in Nature Communications.

1. One of my major concerns of this manuscript is the novelty. The related luminescent mechanism of room temperature phosphorescence, applications and general strategy are almost identical to published works. (Sci. Adv. 2020, 6, eaaz6107; Angew. Chem. 2020, 132, 9479–9483; ACS Appl. Mater. Interfaces 2020, 12, 20765–20774; Adv. Mater. 2020, 32, 1907355)

Reply: Thanks for the comments. Here, we would like to re-clarify the main novelties of this work:

- (1) Facile reaction in pure aqueous phase with $\text{NH}_3 \cdot \text{H}_2\text{O}$ as catalyst to construct B-O covalent bond between DPP-BOH and PVA matrix, which then led to the ultralong RTP lifetime of 2.43 s.
- (2) Adjusted afterglow colors from blue to green to orange through triplet-to-singlet Förster-resonance energy-transfer (TS-FRET) with the introduction of two purely organic fluorescent dyes.
- (3) Reversible stimulus-responsive RTP effect for the colorful afterglow systems.

As for the previous works raised by the Reviewer, they are undoubtedly very beautiful. However, they would not show negative effect on the novelty of this work, as many differences exist between these previous works and this one. Actually, based on many beautiful literatures including the above mentioned ones, in this work, we further achieved some exciting developments. To clearly demonstrate the differences and the developed novelties of our work, we would like to do some comparisons with the mentioned four literatures, however, this does not mean we aim to undermine the beautiful works of other scientists.

Sci. Adv. 2020, 6, eaaz6107: Similar reaction was utilized to construct B-O covalent bond between TPEDB and PVA matrix, but K_2CO_3 and KHCO_3 with metal element were added as catalyst, in which the influence of metal element on RTP behaviors was hard to be clarified. The resultant system only showed green RTP emission with much shorter lifetime of 768.6 ms. Furthermore, no stimulus-responsive RTP effect was reported.

Angew. Chem. 2020, 132, 9479–9483: Triplet-to-singlet Förster-resonance energy-transfer (TS-FRET) was utilized to adjust the afterglow color, but the metal elements also existed in the RTP donor and fluorescent acceptor. No covalent bond between PVA and phosphorescent chromophore was formed and no stimulus-responsive RTP effect

was reported.

ACS Appl. Mater. Interfaces 2020, 12, 20765–20774: No covalent bond between PVA and phosphorescent chromophore was formed, which then resulted in the much shorter RTP lifetimes (less than 0.6 s). Also, just green RTP emission was observed in this work.

Adv. Mater. 2020, 32, 1907355: No covalent bond between PVA and phosphorescent chromophore was formed, which then resulted in the much shorter RTP lifetime of 1.29 s. The RTP colors were just limited in the range of blue to green.

As pointed in the comments of Reviewers 2 and 3, the importance and novelty of this work have been appreciated as “*Room temperature phosphorescent materials are a topic that has attracted a great deal of attention in recent years, and this simple method for creating stimuli-responsive, multi-color phosphorescence materials could be an important contribution.* (Reviewer 2)” and “*The manuscript has enough novelty and is suitable to publish in this journal considering the recent interest in ambient organic phosphors.* (Reviewer 3)”.

2. *The DPP-BOH-PVA construction and the achievement of multiple color afterglow through energy transfer, which is the same as Ref (Sci. Adv. 2020, 6, eaaz6107 and Angew. Chem. 2020, 132, 9479–9483). Therefore, new fundamental insights could be found in this manuscript.*

Reply: Thanks for the comment. As described above, many differences exist between these previous works and this one:

Sci. Adv. 2020, 6, eaaz6107: Facile reaction was utilized to construct B-O covalent bond between TPEDB and PVA matrix, but K_2CO_3 and $KHCO_3$ with metal element were added as catalyst, in which the influence of metal element on RTP behaviors was hard to be clarified. As for our work, the optimized reaction condition was utilized, in which the metal-free $NH_3 \cdot H_2O$ was added as the catalyst instead of the metal ones. Besides, the system based on TPEDB-PVA only show green RTP emission with much shorter lifetime of 768.6 ms. Furthermore, no stimulus-responsive RTP effect was reported.

Angew. Chem. 2020, 132, 9479–9483: Triplet-to-singlet Förster-resonance energy-transfer (TS-FRET) was utilized to adjust the afterglow color, but the metal elements existed in the RTP donor and fluorescent acceptor. Also, the influence of metal element on RTP behaviors was hard to be clarified. Besides, no covalent bond between PVA and phosphorescent chromophore was formed and no stimulus-responsive RTP effect was reported.

Thus, it is the case that some similarities might exist between these previous works and this one, however, the differences are also stark. It is believed that the colorful and unique stimulus-responsive RTP effect in our work would attract a broad interest of readers.

3. *From the introduction, the authors stated that “it is necessary to develop ... another dimension”. It is so unclear to understand this, what is the dimension exactly? It may lead to some ambiguity for the reader.*

Reply: Thanks for the comments. Accordingly, we have changed the statement as

“Therefore, it is necessary to develop stimulus-responsive materials from another dimension, such as emission lifetime, which could broaden their practical application in much more fields.” in the revised manuscript to make a more clear presentation. (Page 3, line3)

4. From the Figure 2d, between the cycle of removal of water and heating, the phosphorescence color showed some obvious changes. For example, the color was green while turned to blue after one cycle of treatment, but it changed to green again after another cycle, which is significant in Supporting Figure 2. What are the internal reasons? Such problems also existed in the Figure 2e, fluorescence color changed under various temperatures. And Supporting Figure 4.

Reply: Thanks for the comments. The changed phosphorescence color should come from the chromatic aberration when the photos were taken, while no changes could be observed for the corresponding RTP spectra. Accordingly, we have re-taken the RTP pictures with iPhone 6 in a more uniform condition and added them in the revised Figure 2 and Supplementary Figure 2 and 4. As we can see, the resultant photos all show similar RTP color.

Fig. 2. Photophysical properties of DPP-BOH-PVA film under the stimuli of water and heat. **a** Phosphorescence spectra of water-fumed DPP-BOH-PVA film after heating at different temperatures for 15 min. **b** Time-resolved emission-decay profiles of water-fumed DPP-BOH-PVA film after heating at different temperatures for 15 min. **c** Phosphorescence spectra of desiccative DPP-BOH-PVA film under water fuming for different times. **d** Repeated cycles of the heating/water fuming processes and the corresponding photographs of DPP-BOH-PVA film after turning off the UV irradiation. **e** Photographs of water-fumed DPP-BOH-PVA film after heating at different temperatures (30 °C-80 °C). The temperature gradient was 10 °C and the corresponding

heating time was 15 min, after which the RTP behaviors were studied when the samples were cooled to room temperature.

Supplementary Figure 2. Photographs of repeated cycles of the heating/water fuming processes for DPP-BOH-PVA film.

Supplementary Figure 4 (Supplementary Figure 7 in the revised Supplementary Information). Photographs of water-fumed DPP-BOH-PVA-C films after heating at different temperatures (30 °C-80 °C). The temperature gradient was 10 °C and the corresponding heating time was 15 min, after which the RTP behaviors were studied when the samples were cooled to room temperature.

5. For the Figure 2d, after just one cycle treatment, the phosphorescence intensity decreased significantly. Whether the high temperature treatment would destroy the internal structures, which need to be verified by experiments.

Reply: Thanks for the comments. To study the influence of heat treatment, the differential scanning calorimetry (DSC) and thermogravimetric analysis (TGA) of PVA, DPP-BOH-PVA, DPP-BOH-PVA-F and DPP-BOH-PVA-R were carried out and the corresponding data have been added in the revised Supplementary Figure 3-4 and Supplementary Table 3. As we can see, the glass transition temperatures and decomposition temperatures of these materials are about 66 °C and 320 °C, respectively. Thus, after heat treatment at 80 °C, materials could just undergo a slight glass transition,

which would not lead to the significantly decreased RTP intensity. It is considered that the minor difference in test conditions, including the standing time of film after heating, should be mainly responsible for the different RTP intensity, as the activated film would absorb moisture from the air again.

In addition, we repeated the cycle experiment in a more uniform condition and the results have been added in the revised Figure 2d, in which no obvious decrease for RTP emission could be found after heat treatment.

Supplementary Figure 26. The differential scanning calorimetry (DSC) curves of PVA, DPP-BOH-PVA, DPP-BOH-PVA-F and DPP-BOH-PVA-R films.

Supplementary Figure 27. The thermogravimetric analysis (TGA) curves of **PVA**, **DPP-BOH-PVA**, **DPP-BOH-PVA-F** and **DPP-BOH-PVA-R** films.

Supplementary Table 4. Glass transition temperatures and decomposition temperatures of **PVA**, **DPP-BOH-PVA**, **DPP-BOH-PVA-F** and **DPP-BOH-PVA-R** films.

	PVA	DPP-BOH-PVA	DPP-BOH-PVA-F	DPP-BOH-PVA-R
Glass transition temperature/°C	59.3	66.5	66.7	66.8
Decomposition temperature/°C	308.5	308.5	324.6	323.7

Figure 2d Repeated cycles of the heating/water fuming processes and the corresponding photographs of **DPP-BOH-PVA** film after turning off the UV irradiation.

6. The authors attributed the new absorption peak to the B-O covalent bond. However, no direct evidence can be derived from this manuscript to verify this. Some additional characterizations should be presented.

Reply: Thanks for the comments. In order to study the reaction between DPP-BOH and PVA and the yield of this reaction, ¹H NMR and X-ray photoelectron spectroscopy (XPS) were carried out firstly. However, because the concentration of DPP-BOH in the PVA film is too low (~1%, wt%), the sensitivity of these two methods is not sufficient for analysis, as shown in Reply Figure 1-3.

Then, we tried to prove it by UV absorption. DPP-BOH-PVA and DPP-BO-PVA prepared aqueous solutions were extracted by dichloromethane (DCM) with the volume ratio of 1:1, then their UV-vis absorption spectra in aqueous phase and organic phase were measured respectively. Comparing the UV absorption spectra of DPP-BOH-PVA and DPP-BO-PVA solutions in Supplementary Figure 5, it could be found there is still obvious absorption in the aqueous phase after DCM extraction for DPP-BOH-PVA, while nearly no absorption could be observed in the aqueous phase after DCM extraction for DPP-BO-PVA. This indicates strong interactions, such as B-O covalent

bond between PVA and DPP-BOH, have been formed, which makes the phosphorescent chromophore hard to be extracted by DCM solution. According to Beer-Lambert law, we can roughly calculate the reaction yield of 46.50% for DPP-BOH-PVA.

Moreover, the reaction between boric acid derivatives and organic sugars or polyol compounds to form boronic ester with structural rigidity have been reported, such as J. Org. Chem. 1959, 24, 769–774; Adv. Funct. Mater. 2019, 29, 1905514, Sci. Adv. 2020, 6, eaaz6107, J. Am. Chem. Soc. 2011, 133, 660–663 and Nat. Chem. 2014, 6, 1003–1008. These previous works indicated that the reaction between boric acid and PVA is very mature. Besides, none of these works have provided detailed proof for the occurrence of this reaction.

The reaction yield was calculated based on following equation:

$$A = \lg(1/T) = Kbc \quad (a)$$

Where A, T, K, b and c are absorbance, transmittance, molar absorption coefficient, the distance the light travels in the sample and the concentration of the solution. The absorbance of aqueous phase and DCM solution after extraction of DPP-BOH-PVA aqueous solution is 0.113 and 0.130, respectively.

The absorption spectra and calculation equations have been added in revised Supplementary Figure 5.

Also, “The reaction yield was proved to be 46.50% by UV absorption measurement (Supplementary Fig. S5)” has been added in the revised manuscript. (Page 7, line 7)

Reply Figure 1. ^1H NMR spectrum of DPP-BOH-PVA in $\text{DMSO-}d_6$ at room temperature.

Reply Figure 2. The survey and C 1s, O 1s and B 1s high-resolution X-ray photoelectron spectroscopy (XPS) spectra of **DPP-BOH-PVA**.

Reply Figure 3. The survey and C 1s, O 1s and B 1s high-resolution X-ray photoelectron spectroscopy (XPS) spectra of **DPP-BOH-PVA-C**.

Supplementary Figure 5. The UV-vis absorption spectra of aqueous phase (green line) and dichloromethane (DCM) solution (red line) for a) **DPP-BOH-PVA** and b) **DPP-BO-PVA** after extracting their prepared aqueous solutions by dichloromethane (DCM) with the volume ratio of 1:1.

7. *The proposed large ISC and low irradiative constant should be presented as well.*

Reply: Thanks for the comments. Accordingly, we calculated the rate constant of intersystem crossing (k_{isc}) and phosphorescence irradiative rate ($k_{p,nr}$) based on the equations below:

$$k_{f,r} = \Phi_f / \tau_f \quad (b)$$

$$\Phi_{isc} = 1 - \Phi_f - \Phi_{ic} \approx 1 - \Phi_f \quad (c)$$

$$\tau_f = 1 / (k_{f,r} + k_{f,nr} + k_{isc});$$

$$\Phi_{isc} = k_{isc} / (k_{f,r} + k_{f,nr} + k_{isc}) = k_{isc} \times \tau_f;$$

$$k_{isc} = \Phi_{isc} / \tau_f \quad (d)$$

$$\tau_p = 1 / (k_{p,r} + k_{p,nr});$$

$$\Phi_p = (\Phi_{isc} \times k_{p,r}) / (k_{p,r} + k_{p,nr}) = \Phi_{isc} \times k_{p,r} \times \tau_p;$$

$$k_{p,r} = \Phi_p / (\Phi_{isc} \times \tau_p) \quad (e)$$

$$k_{p,nr} = 1 / \tau_p - k_{p,r} \quad (f)$$

Where, $k_{f,r}$, k_{isc} , $k_{p,r}$, $k_{p,nr}$ are the radiative rate constant of prompt fluorescence, rate constant of intersystem crossing (ISC), radiative rate constant of phosphorescence and non-radiative rate constant of phosphorescence.

For DPP-BOH-PVA, $k_{isc} = 6.34 \times 10^7 \text{ s}^{-1}$, $k_{p,nr} = 0.37 \text{ s}^{-1}$

The calculation results and equations have been added in the revised Supplementary Table 5. As we can see, the k_{isc} is efficient enough for the generation of triplet excitons and high RTP efficiency (7.51%), while $k_{p,r}$ and $k_{p,nr}$ are small enough for realizing the ultralong RTP lifetime (2.43 s) of DPP-BOH-PVA. Also, the related discussions have been added in the revised manuscript as “Besides, the k_{ISC} and $k_{P,r}$ for DPP-BOH-PVA film were calculated based on the experimental results. As shown in Supplementary Table 5, the k_{isc} is as large as $6.34 \times 10^7 \text{ s}^{-1}$, while $k_{p,r}$ and $k_{p,nr}$ are just 0.04 and 0.37 s^{-1} , which could well correspond to the theoretical results, and certify the accuracy of internal mechanisms mentioned above.” (Page 8, line 14)

Supplementary Table 5. Dynamic photophysical parameters of DPP-BOH-PVA.

τ_f/ns	τ_p/s	$\Phi_f/\%$	$\Phi_p/\%$	$k_{r,r}/\text{s}^{-1}$	$\Phi_{isc}/\%$	k_{isc}/s^{-1}	$k_{p,r}/\text{s}^{-1}$	$k_{p,nr}/\text{s}^{-1}$	f
12.99	2.43	17.60	7.51	1.35×10^7	82.40	6.34×10^7	0.04	0.37	0.0078

8. Such stimulus-responsive characteristics were just from the PVA, which can absorb water from atmosphere. Hence, such materials can be really called smart responsive materials? Additionally, the quenching of phosphorescence by water in PVA matrix have been reported in some works (*ACS Appl. Mater. Interfaces* 2020, 12, 20765–20774; *Adv. Mater.* 2020, 32, 1907355).

Reply: Thanks for the comments. As described in the comment of Reviewer 2 “Room temperature phosphorescent materials are a topic that has attracted a great deal of attention in recent years, and this simple method for creating stimuli-responsive, multi-color phosphorescence materials could be an important contribution”. By utilizing the sensitivity of PVA to water to construct stimulus-responsive RTP system is very simple, however, it indeed provides an efficient and universal way to realize multi-color stimulus-responsive RTP effect, which has been well demonstrated in this work.

In the previous work of *ACS Appl. Mater. Interfaces* 2020, 12, 20765–20774, it mainly utilized the intermolecular hydrogen bonds between phosphorescent chromophores and PVA to construct RTP materials with the longest lifetime of 0.585 s in green emission. As for the work of *Adv. Mater.* 2020, 32, 1907355, two polyphosphazene derivatives containing carbazolyl units were doped into poly(vinyl alcohol) (PVA) films, the excitation-dependent RTP ranging from blue to green could be observed, with lifetime of 1.29 s. Since PVA acted as polymer matrix in these two works, the water/heat-responsive RTP phenomena could be observed for them, although no detailed studies were carried out.

In our work, the facile reaction in pure aqueous phase with $\text{NH}_3 \cdot \text{H}_2\text{O}$ as catalyst was utilized to construct B-O covalent bond between DPP-BOH and PVA matrix, which led to the ultralong RTP lifetime of 2.43 s. Then, with the introduction of two purely organic fluorescent dyes, the adjusted afterglow colors from blue to green to orange through triplet-to-singlet Förster-resonance energy-transfer (TS-FRET) were realized. Also, reversible stimulus-responsive RTP effect with colorful afterglow emissions was observed for the existing of PVA matrix.

A careful comparison between the two previous works and this one, it can be found that some similarities might exist between these previous works and this one, however, the differences are also stark. Particularly, the ultralong RTP lifetime (2.43 s) and wide RTP emission colors (blue-green-yellow) in this work would undoubtedly promote their potential applications and attract wider interest from readers.

9. For the application as shown in Figure 5, the fingerprint displays were too vague to identify, which should be improved.

Reply: Thanks for the comments. Accordingly, we have improved the fingerprint displays and added it in the revised Figure 5d.

Fig. 5. Applications of stimulus-responsive afterglow materials. The schematic illustration of application process for **a** flexible film, **b** anti-counterfeit, **c** screen printing and **d** fingerprint record.

Reviewer #2 (Remarks to the Author):

The authors report a new organic room temperature phosphorescent material that responds to external stimuli. A blue room temperature phosphorescent material is obtained by the simple method of mixing poly(vinyl alcohol) (PVA) and a phosphorescent dye containing boronic acid in water and allowing it to dry. The importance of hydrogen bond formation in PVA for the emission of room temperature phosphorescence is investigated by several controlled experiments, and the addition of water breaks the hydrogen bonds, resulting in the switch-off of the phosphorescence. Demonstrations of multicolor phosphorescence and anti-counterfeiting applications show the usefulness of the current materials. Room temperature phosphorescent materials are a topic that has attracted a great deal of attention in recent years, and this simple method for creating stimuli-responsive, multi-color phosphorescence materials could be an important contribution. The following points should be considered before publishing this paper.

1. The NTOs of T_m involved in the inter-system crossing from S_1 to should be shown to see if there is any change in the shape of the orbitals.

Reply: Thanks for the comments. Accordingly, we have calculated the NTOs of T_m involved in the inter-system crossing from S_1 and added them in Supporting Figure 29. Also, the spin orbit coupling (SOC) constants of $T_m \rightarrow S_1$ and $T_1 \rightarrow S_0$ were calculated and added in Supplementary Figure 15-16 of the revised Supplementary Information.

As we can see, the T_m states with charge transfer from hole to particle would give larger SOC constants between S_1 , thus much benefiting for the generation of triplet excitons and high RTP efficiency (7.51%). Besides, the SOC value from T_1 to S_0 is relatively small for the LE (local excited) character of T_1 state, thus leading to the much slower phosphorescence radiative rate and ultra-long phosphorescence lifetime (2.43 s).

Supplementary Figure 15. The theoretical calculations about natural transition orbitals (NTOs) for DPP-BOH.

Supplementary Figure 16. Calculated energy diagram and spin-orbit coupling (SOC) value (ξ) of DPP-BOH.

2. The efficiency of ISC is desired to be determined.

Reply: Thanks for the comment. Accordingly, we calculated the rate constant of intersystem crossing (k_{isc}) and efficiency of ISC (Φ_{isc}) for DPP-BOH-PVA based on the equations below:

$$k_{f,r} = \Phi_f / \tau_f \quad (b)$$

$$\Phi_{isc} = 1 - \Phi_f - \Phi_{ic} \approx 1 - \Phi_f \quad (c)$$

$$\tau_f = 1 / (k_{f,r} + k_{f,nr} + k_{isc});$$

$$\Phi_{isc} = k_{isc} / (k_{f,r} + k_{f,nr} + k_{isc}) = k_{isc} \times \tau_f;$$

$$k_{isc} = \Phi_{isc} / \tau_f \quad (d)$$

$$\tau_p = 1 / (k_{p,r} + k_{p,nr});$$

$$\Phi_p = (\Phi_{isc} \times k_{p,r}) / (k_{p,r} + k_{p,nr}) = \Phi_{isc} \times k_{p,r} \times \tau_p;$$

$$k_{p,r} = \Phi_p / (\Phi_{isc} \times \tau_p) \quad (e)$$

$$k_{p,nr} = 1 / \tau_p - k_{p,r} \quad (f)$$

Where, $k_{f,r}$, k_{isc} , $k_{p,r}$, $k_{p,nr}$ are the radiative rate constant of prompt fluorescence, rate constant of intersystem crossing (ISC), radiative rate constant of phosphorescence and non-radiative rate constant of phosphorescence.

Also, the related data have added in the Supplementary Table 5 of revised Supplementary Information. As we can see, the Φ_{isc} (82.40%) and k_{isc} ($6.34 \times 10^7 \text{ s}^{-1}$) are large enough for the generation of triplet excitons and high RTP efficiency (7.51%).

Supplementary Table 5. Dynamic photophysical parameters of ultralong organic phosphorescence.

τ_f/ns	τ_p/s	$\Phi_f/\%$	$\Phi_p/\%$	$k_{f,r}/\text{s}^{-1}$	$\Phi_{isc}/\%$	k_{isc}/s^{-1}	$k_{p,r}/\text{s}^{-1}$	$k_{p,nr}/\text{s}^{-1}$	f
12.99	2.43	17.60	7.51	1.35×10^7	82.40	6.34×10^7	0.04	0.37	0.0078

3. For FRET from T_1 to fluorescein or rhodamine in DPP-BOH-PVA, the validity of FRET occurring from T_1 with a small transition moment should be discussed by calculating the rate constant of FRET and comparing it with the deactivation rate of the triplet.

Reply: Thanks for the comments. Accordingly, the rate constant of FRET was calculated using the following equations:

$$k_{ET}(R) = (R_0/R)^6 / \tau_{0,D} \quad (i)$$

$$E = R_0^6 / (R_0^6 + R^6) \quad (j)$$

Where k_{ET} , $\tau_{0,D}$, E , R and R_0 are the rate constant of FRET, the lifetime of donor without acceptor, the efficiency of FRET, the transfer radius and the critical transfer radius where FRET occurs effectively.

The rate constants of FRET from DPP-BOH-PVA to fluorescein and rhodamine B were calculated to be 0.06 s^{-1} and 0.04 s^{-1} respectively. As the rate constant (0.06 s^{-1}) of FRET from DPP-BOH-PVA to fluorescein is larger than the RTP radiative rate ($k_{p,r} = 0.04 \text{ s}^{-1}$) of DPP-BOH-PVA, the triplet-to-singlet Förster-resonance energy-transfer (TS-FRET) between DPP-BOH-PVA and fluorescein occurred with a relatively high efficiency of 13.58%. Besides, the rate constant (0.04 s^{-1}) of FRET from DPP-BOH-PVA to rhodamine B is similar to the RTP radiative rate ($k_{p,r} = 0.04 \text{ s}^{-1}$) of DPP-BOH-PVA, thus the TS-FRET between DPP-BOH-PVA and rhodamine B occurred in a relatively low efficiency of 9.47%. Also, the related discussions have been added in the revised manuscript as “Besides, the rate constants of FRET from **DPP-BOH-PVA** to fluorescein and rhodamine B were calculated to be 0.06 s^{-1} and 0.04 s^{-1} respectively. As these data were larger or similar to the RTP radiative rate (0.04 s^{-1}) of **DPP-BOH-PVA**, the FRET occurring from singlet to triplet state could be well certified (Supplementary Table 6).”. (Page 11, line 1)

In addition, the calculation results and equations have been added in Supplementary Table 6. The radiative rate constant of phosphorescence ($k_{p,r}$) of DPP-BOH-PVA has been calculated and been added in Supplementary Table 5.

Supplementary Table 6. Calculated rate constant and efficiency of FRET from **DPP-BOH-PVA** to fluorescein and rhodamine B.

	$E_{S \rightarrow S}/\%$	$E_{T \rightarrow S}/\%$	$\tau_{0,D}/\text{s}$	k_{ET}/s^{-1}
DPP-BOH-PVA-F	12.86	13.58	2.43	0.06
DPP-BOH-PVA-R	13.32	9.47		0.04

Reviewer #3 (Remarks to the Author):

In the manuscript the authors have demonstrated stimulus-responsive cyan emissive room temperature phosphorescence from aryl boronic acid chromophores in PVA with an afterglow emission. Authors have explained that the formation of B-O covalent bond between phosphor and chromophores is responsible for the phosphor's increased rigidity, which results in a high phosphorescence lifetime. By taking advantage of the afterglow emission of the phosphors, authors have demonstrated green and orange afterglow emission from conventional fluorescence emitter through phosphorescence Förster resonance energy transfer (P-FRET/TS-FRET) process. On the other hand, very efficient phosphorescence emission in the low wavelength region with a lifetime of 2.43 s is another important aspect of this manuscript. Also, the authors have demonstrated multicolor security printing, which is beneficial for multilevel information encryption. The manuscript has enough novelty and is suitable to publish in this journal considering the recent interest in ambient organic phosphors. However, the author should address the following points in a revised manuscript, to quantify certain aspects which is important for a publication in a journal like Nature Communication.

1. To achieve high phosphorescence efficiency and lifetime, authors have heated the films to 80 °C. What will happen if authors go more than 80 °C? As I can see that emission is saturated at 60°C, but lifetime is still increasing.

Reply: Thanks for the comments. Firstly, we measured the differential scanning calorimetry (DSC) and thermogravimetric analysis (TGA) of PVA, DPP-BOH-PVA, DPP-BOH-PVA-F and DPP-BOH-PVA-R, and the related data have been added in Supplementary Figure 3-4 and Supplementary Table 3. As we can see, the glass transition temperatures and decomposition temperatures of these materials are about 66 °C and 320 °C, respectively. Then, if heat the film more than 66 °C, materials would undergo glass transition, which gives a negative influence on the resultant RTP intensity and lifetime. On the other hand, to remove the moisture from the film, the higher temperature will be better. Thus, the influence of high temperature (> 66 °C) on RTP would be hard to be predicted.

To study it, the RTP spectra and phosphorescence lifetimes of water-fumed DPP-BOH-PVA film after heating from 30 °C-130 °C were measured again. At this time, the RTP intensity and lifetime are both saturated at about 80 °C. However, if the heating temperature was more than 100 °C, the resultant RTP lifetime would show some decreasing. To sum up, 80 °C should be an appropriate heating temperature to activate RTP effect of these films.

Supplementary Figure 3. The differential scanning calorimetry (DSC) curves of PVA, DPP-BOH-PVA, DPP-BOH-PVA-F and DPP-BOH-PVA-R films.

Supplementary Figure 4. The thermo gravimetric analysis (TGA) curves of PVA, DPP-BOH-PVA, DPP-BOH-PVA-F and DPP-BOH-PVA-R films.

Supplementary Table 3. Glass transition temperature and decomposition temperature of PVA, DPP-BOH-PVA, DPP-BOH-PVA-F and DPP-BOH-PVA-R films.

	PVA	DPP-BOH-PVA	DPP-BOH-PVA-F	DPP-BOH-PVA-R
Glass transition temperature/°C	59.3	66.5	66.7	66.8
Decomposition temperature/°C	308.5	308.5	324.6	323.7

Reply Figure 4. **a** Phosphorescence spectra of water-fumed **DPP-BOH-PVA** film after heating at different temperatures for 15 min. **b** Time-resolved emission-decay profiles of water-fumed **DPP-BOH-PVA** film after heating at different temperatures for 15 min. **c** Changes of phosphorescence intensity with temperature increasing for water-fumed **DPP-BOH-PVA** film. **d** Changes of phosphorescence lifetime with temperature increasing for water-fumed **DPP-BOH-PVA** film.

Reply Figure 5. Photographs of water-fumed DPP-BOH-PVA film after heating at different temperatures (30 °C-130 °C). The temperature gradient was 10 °C and the corresponding heating time was 15 min, after which the RTP behaviors were studied when the samples were cooled to room temperature.

Reply Table 1. Phosphorescence lifetimes of water-fumed DPP-BOH-PVA, DPP-BOH-PVA-F and DPP-BOH-PVA-R film after heating at different temperatures for 15 min for first measurement.

	DPP-BOH-PVA	DPP-BOH-PVA-F	DPP-BOH-PVA-R
	τ_p (s) (475 nm)	τ_p (s) (475 nm, 533 nm)	τ_p (s) (475 nm, 581 nm)
H ₂ O	0.01	0.01, 0.01	0.01, 0.01
30 °C	0.01	0.01, 0.01	0.01, 0.01
40 °C	0.88	0.61, 0.51	1.20, 0.89
50 °C	1.37	0.90, 0.69	1.60, 1.22
60 °C	2.13	1.48, 1.18	1.87, 1.44
70 °C	2.57	1.83, 1.49	2.19, 1.73
80 °C	2.69	2.03, 1.70	2.32, 1.87
90 °C	2.82	2.28, 1.93	2.49, 2.02
100 °C	2.85	2.38, 2.06	2.54, 2.07
110 °C	2.83	2.43, 2.13	2.55, 2.09
120 °C	2.82	2.45, 2.20	2.54, 2.06

130 °C	2.78	2.47, 2.25	2.50, 2.01
--------	------	------------	------------

2. Is there any experimental support that can quantitatively tell the conversion of B-O covalent bond formation.

Reply: Thanks for the comment. In order to study the reaction yield between DPP-BOH and PVA, ¹H NMR and X-ray photoelectron spectroscopy (XPS) were carried out firstly. However, because the concentration of DPP-BOH in the PVA film is too low (~1%, wt%), the sensitivity of these two methods is not sufficient for analysis, as shown in Reply Figure 1-3.

Then, we tried to prove it by UV absorption. DPP-BOH-PVA and DPP-BO-PVA prepared aqueous solutions were extracted by dichloromethane (DCM) with the volume ratio of 1:1, then their UV-vis absorption spectra in aqueous phase and organic phase were measured respectively. Comparing the UV absorption spectra of DPP-BOH-PVA and DPP-BO-PVA solutions in Supplementary Figure 4, it could be found there is still obvious absorption in the aqueous phase after DCM extraction for DPP-BOH-PVA, while nearly no absorption could be observed in the aqueous phase after DCM extraction for DPP-BO-PVA. This indicates strong interactions, such as B-O covalent bond between PVA and DPP-BOH, have been formed, which makes the phosphorescent chromophore hard to be extracted by DCM solution. According to Beer-Lambert law, we can roughly calculate the reaction yield of 46.50%.

The reaction yield was calculated based on following equation:

$$A = \lg(1/T) = Kbc \quad (a)$$

Where A, T, K, b and c are absorbance, transmittance, molar absorption coefficient, the distance the light travels in the sample and the concentration of the solution. The absorbance of aqueous phase and DCM solution after extraction of DPP-BOH-PVA is 0.113 and 0.130, respectively.

The absorption spectra and calculation equations have been added in revised Supporting Figure 28.

Also, “The reaction yield was proved to be 46.50% by UV absorption measurement (Supplementary Fig. 5)” has been added in the revised manuscript. (Page 7, line 7)

Reply Figure 1. ^1H NMR spectrum of DPP-BOH-PVA in $\text{DMSO-}d_6$ at room temperature.

Reply Figure 2. The survey and C 1s, O 1s and B 1s high-resolution X-ray photoelectron spectroscopy (XPS) spectra of DPP-BOH-PVA.

Reply Figure 3. The survey and C 1s, O 1s and B 1s high-resolution X-ray photoelectron spectroscopy (XPS) spectra of DPP-BOH-PVA-C.

Supplementary Figure 5. The UV-vis absorption spectra of aqueous phase (green line) and dichloromethane (DCM) solution (red line) for a) DPP-BOH-PVA and b) DPP-BO-PVA after extracting their prepared aqueous solutions by dichloromethane (DCM) with the volume ratio of 1:1.

3. *Cyan emissive persistent phosphorescence emission is one of the exciting parts of this manuscript; authors should cite similar recent work, which is very relevant for this manuscript Adv. Funct. Mater. 2020, 30, 2003693.*

Reply: Thanks for your comment. Accordingly, this reference has been cited as ref 18 in the revised manuscript.

4. In the Supplementary Table 1, I can see no decrease in the donor lifetime at 30 °C, 60 °C, and 70 °C with acceptor undoped and doped samples. Is FRET is not operating here?

Reply: Thanks for the comments. As shown in the Supplementary Table 1, the emission peaks at 533 nm (fluorescein) and 581 nm (rhodamine B) could show ultralong lifetimes even more than 1s at 60 °C and 70 °C, indicating the triplet-to-singlet FRET should have occurred. It is considered that the minor difference in test conditions, such as atmospheric temperature/humidity and standing time of film after heating and even film preparation, should be mainly responsible for the imperfect data in Supplementary Table 1. On one hand, the activated film after heating would re-absorb water vapor in the air, thus the minor difference in atmospheric temperature/humidity and standing time of film after heating could largely affect the resultant RTP behaviors; on the other hand, the film state, especially for the non-uniformity (Reply Figure 6), would also lead to the much difference in the resultant RTP data.

Reply Figure 6. Photographs under optical microscope (up) and 3D laser confocal microscope (down) of a) DPP-BOH-PVA, b) DPP-BOH-PVA-F and c) DPP-BOH-PVA-R film.

In addition, we re-prepared the films of DPP-BOH-PVA, DPP-BOH-PVA-F and DPP-BOH-PVA-R, then measured the corresponding RTP lifetimes for another two times in the more uniform conditions (Reply Table 2-3). Really, the random variations still existed in the RTP behaviors after heating. However, the careful analysis demonstrated that both of the new experimental results and those presented in the origin version of our manuscript, gave the same overall trend (a and b). Thus, the results are repeatable regardless of the unvoided disturbings, once again confirming the explanations presented in the text.

- (a) The RTP lifetime for each film increased with the raised heating temperature and was saturated at about 80 °C.
- (b) The triplet-to-singlet FRET efficiency from DPP-BOH to fluorescein (12.8% or 16.5%) is larger than that (5.4% or 4.1%) from DPP-BOH to rhodamine B at 80 °C.

Reply Table 2. Phosphorescence lifetimes of water-fumed **DPP-BOH-PVA**, **DPP-BOH-PVA-F** and **DPP-BOH-PVA-R** films after heating at different temperatures for 15 min for second measurement.

	DPP-BOH-PVA	DPP-BOH-PVA-F	DPP-BOH-PVA-R
	τ_P (s) (475 nm)	τ_P (s) (475 nm, 533 nm)	τ_P (s) (475 nm, 581 nm)
H ₂ O	0.01	0.01, 0.01	0.01, 0.01
30 °C	0.88	0.01, 0.01	0.01, 0.01
40 °C	1.91	0.98, 0.60	0.91, 0.72
50 °C	1.98	1.66, 1.19	1.54, 1.03
60 °C	2.24	2.05, 1.76	2.21, 1.74
70 °C	2.51	2.24, 1.95	2.46, 1.94
80 °C	2.58	2.25, 2.08	2.44, 2.06
90 °C	2.62	2.36, 2.13	2.58, 2.08
100 °C	2.70	2.38, 2.16	2.59, 2.11
110 °C	2.67	2.43, 2.23	2.61, 2.11
120 °C	2.70	2.43, 2.22	2.48, 2.05
130 °C	2.67	2.46, 2.25	2.44, 2.06

Reply Table 3. Phosphorescence lifetimes of water-fumed **DPP-BOH-PVA**, **DPP-BOH-PVA-F** and **DPP-BOH-PVA-R** films after heating at different temperatures for 15 min for third measurement.

	DPP-BOH-PVA	DPP-BOH-PVA-F	DPP-BOH-PVA-R
	τ_P (s) (475 nm)	τ_P (s) (475 nm, 533 nm)	τ_P (s) (475 nm, 581 nm)
H ₂ O	0.01	0.01, 0.01	0.01, 0.01
30 °C	0.01	0.01, 0.01	0.01, 0.01
40 °C	1.46	0.62, 0.43	1.36, 0.98
50 °C	1.74	1.26, 0.96	1.60, 1.21
60 °C	2.17	1.68, 1.30	1.89, 1.50
70 °C	2.44	1.98, 1.58	2.15, 1.77
80 °C	2.66	2.22, 1.81	2.55, 2.09
90 °C	2.90	2.30, 1.93	2.60, 2.20
100 °C	2.85	2.37, 2.04	2.63, 2.21
110 °C	2.85	2.51, 2.22	2.68, 2.25
120 °C	2.86	2.50, 2.22	2.67, 2.25

5. To get the delayed fluorescence lifetime from Fluorescein and Rhodamine B, authors have monitored the emission wavelength at 533 nm 581 nm; however, a significant donor contribution is observed. How did the author nullify the contribution from the donor lifetime in the delayed fluorescence lifetime of fluorescein and rhodamine B?

Reply: Thanks for the comments. Accordingly, the time-resolved phosphorescence spectra after turning off the UV lamp were measured and have been added in Supplementary Figure 20. As we can see, even after turning off the UV lamp for 9 seconds, the peaks at 533 nm (fluorescein) and 581 nm (rhodamine B) is still visible and the intensity is higher than the peak at 475 nm (DPP-BOH). Therefore, the donor contribution might exist, but is considered negligible.

Supplementary Figure 20. Time-resolved phosphorescence spectra after turning off the UV lamp of a) DPP-BOH-PVA, b) DPP-BOH-PVA-F and c) DPP-BOH-PVA-R film.

6. It is exciting to know whether the acceptors' emission amplification is there or not. On the other hand, I will suggest the authors show the acceptor's emission spectra upon direct excitation of the acceptors.

Reply: Thanks for the comments. Accordingly, the excitation spectra of fluorescein-PVA and rhodamine B-PVA were measured (Reply Figure 7). The optimal excitation wavelengths of fluorescein-PVA and rhodamine B-PVA are 470 nm and 530 nm, respectively. Then, we measured the PL quantum yields of fluorescein-PVA and rhodamine B-PVA with their optimal excitation wavelengths. It is found they are 65.63% and 95.78% respectively, which is higher than DPP-BOH-PVA-F and DPP-BOH-PVA-R. Thus, the acceptors' emission amplification is not there.

Also, the delayed emission spectra of DPP-BOH-PVA-F and DPP-BOH-PVA-R upon excitations of 470 nm and 530 nm were measured respectively. As shown in Supplementary Figure 22, no delayed emission signals could be detected, indicating no RTP effect upon direct excitation of the acceptors.

Reply Figure 7. Excitation spectra of a) fluorescein in PVA film and b) rhodamine B in PVA film.

Supplementary Figure 22. Phosphorescence spectra of a) DPP-BOH-PVA-F film and b) DPP-BOH-PVA-R film at different excitation wavelengths.

7. I will also suggest that the authors calculate the singlet to singlet FRET efficiency as I can see the decreased donor fluorescence lifetime data from 12.99 ns in the undoped dye sample (supplementary Figure 1) to 11.32 ns 11.76 ns in the fluorescein and rhodamine B (supplementary Figure 13) doped sample.

Reply: Thanks for the comment. The singlet to singlet FRET efficiencies were calculated to be 12.86% and 13.32% for DPP-BOH-PVA-F and DPP-BOH-PVA-R based on the equation of $\Phi_{\text{FRET}} = 1 - \tau_f / \tau_{0,f}$, in which τ_f and $\tau_{0,f}$ are the fluorescence lifetimes of energy donor (DPP-BOH-PVA) after and before energy transfer. The calculated results have been added in Supplementary Table 6.

Supplementary Table 6. Calculated rate constant and efficiency of FRET from DPP-BOH-PVA to fluorescein and rhodamine B.

	$E_{S \rightarrow S} / \%$	$E_{T \rightarrow S} / \%$	$\tau_{0,D} / \text{s}$	$k_{\text{ET}} / \text{s}^{-1}$
DPP-BOH-PVA-F	12.86	13.58	2.43	0.06
DPP-BOH-PVA-R	13.32	9.47		0.04

8. Other than the spectral overlap of the donor emission and the acceptor absorption, significant oscillator strength for the donor phosphorescence is required to facilitate

the dipole-dipole coupling with an acceptor. The author should add more discussion about it.

Reply: Thanks for the comments. According to the equations below, the oscillator strength for the donor phosphorescence was calculated to be $f = 0.0078$, which should be large enough to facilitate the dipole-dipole coupling with an acceptor.

$$k_r = fE_{vert}^2/1.499 \quad (g)$$

$$E_{vert} = 1240/\lambda_p \quad (h)$$

where f , E_{vert} , k_r , λ_p are the oscillator strength, vertical excitation energy, spontaneous radiation rate and first emission peak.

The calculated result has been added in the Supplementary Table 5. Also, the related discussions were added in the revised manuscript as “*Furthermore, the oscillator strength (f) for the donor phosphorescence was calculated to be 0.0078 (Supplementary Table 5), which should be large enough to facilitate the dipole-dipole coupling with an acceptor.*” (Page 11, line 5)

Supplementary Table 5. Dynamic photophysical parameters of ultralong organic phosphorescence.

τ_f /ns	τ_p /s	Φ_f %	Φ_p %	$k_{r,r}$ /s ⁻¹	Φ_{isc} %	k_{isc} /s ⁻¹	$k_{p,r}$ /s ⁻¹	$k_{p,nr}$ /s ⁻¹	f
12.99	2.43	17.60	7.51	1.35×10^7	82.40	6.34×10^7	0.04	0.37	0.0078

9. *Authors should cite the very relevant and seminal research works on TS-FRET process in the phosphorescence energy transfer: J. Phys. Chem. Lett. 2019, 10, 310–315, 2. Sci. Adv. 2019, 5, eaaw5978, Angew. Chem. Int. Ed. 2021, 60, 19720-19724.*

Reply: Thanks for the comment. Accordingly, these references have been cited in ref 28-30.

REVIEWERS' COMMENTS

Reviewer #1 (Remarks to the Author):

In the revision, Li and coauthors have given satisfactory answers to my questions. Therefore, I recommend it to be published in Nat Commun at current stage.

Reviewer #2 (Remarks to the Author):

The authors properly addressed all of my comments. I support the publication of this paper in its current form.

Reviewer #3 (Remarks to the Author):

I appreciate the efforts from the authors to address all the comments and hence i recommend the acceptance of this manuscript in the present form.

REVIEWERS' COMMENTS

Reviewer #1 (Remarks to the Author):

In the revision, Li and coauthors have given satisfactory answers to my questions. Therefore, I recommend it to be published in Nat Commun at current stage.

Reply: Thanks!

Reviewer #2 (Remarks to the Author):

The authors properly addressed all of my comments. I support the publication of this paper in its current form.

Reply: Thanks!

Reviewer #3 (Remarks to the Author):

I appreciate the efforts from the authors to address all the comments and hence i recommend the acceptance of this manuscript in the present form.

Reply: Thanks!